# The Benefits of Radical Treatments with Synchronous Splenectomy for Patients with Hepatocellular Carcinoma and Portal Hypertension

**DOI:** 10.3390/cancers14133155

**Published:** 2022-06-28

**Authors:** Qikun Zhang, Qi Li, Fuchao Shang, Guangming Li, Menglong Wang

**Affiliations:** 1Department of General Surgical Center, Beijing Youan Hospital, Capital Medical University, 8 Xitoutiao, Youwai Street, Fengtai District, Beijing 100069, China; zqkxzmc@ccmu.edu.cn; 2Department of Gastroenterology and Hepatology, Beijing Youan Hospital, Capital Medical University, 8 Xitoutiao, Youwai Street, Fengtai District, Beijing 100069, China; qi.li@ccmu.edu.cn; 3Department of Hepatobiliary and Pancreatic Surgery, The First Hospital of Hebei Medical University, 89 Donggang Street, Shijiazhuang 050031, China; shangfuchao5210@163.com

**Keywords:** synchronous splenectomy, radical treatment, primary hepatocellular carcinoma, portal hypertension

## Abstract

**Simple Summary:**

Radical treatment combined with synchronous splenectomy has recently emerged as an effective therapy for patients with hepatocellular carcinoma (HCC), in the setting of portal hypertension secondary to liver cirrhosis, but its survival benefits remain to be elucidated. We retrospectively analyzed a longitudinal cohort of 96 patients receiving HCC radical treatment combined with splenectomy and a control group comprising 42 patients receiving radical treatment alone, comparing the oncological outcomes of the synchronous splenectomy for the two subgroups. Our analysis highlighted better recurrence-free survival (RFS), particularly in stage T1 patients. Cox multivariate analysis showed that preoperative irregular anti-viral therapy, Child-Pugh grade B liver function, vascular invasion, and microvascular invasion (MVI) were independent risk factors for early postoperative RFS (within 2 years), and preoperative irregular anti-viral therapy and vascular invasion were independent risk factors for 5-year overall survival (OS).

**Abstract:**

***Background:*** The survival benefits of radical treatment (resection or radiofrequency ablation) combined with splenectomy for primary hepatocellular carcinoma (HCC) in patients with liver-cirrhosis-associated portal hypertension (PH) remain to be clarified. ***Methods:*** 96 patients undertaking HCC radical treatment combined with splenectomy (HS group) were retrospectively analyzed, 48 of whom belonged to HCC stage T1 (HSS group). Another 42 patients at stage T1 with PH who received hepatectomy (or radiofrequency ablation) alone (HA group) during the same period served as the control group. Recurrence-free survival (RFS) and overall survival (OS) were compared at each time point between the HSS and HA group. The risk factors affecting early RFS and OS were confirmed through COX multivariate analysis. ***Results:*** The median RFS was 22.3 months and the mean median OS was 46 months in the HS group. As such, 1-year, 2-year, 3-year, and 5-year RFS rates in the HSS and HA group were 95% and 81% (*p* = 0.041), 81% and 67% (*p* = 0.05), 64% and 62% (*p* = 1.00), and 29% and 45% (*p* = 0.10), respectively. Further, 1-year, 3-year, and 5-year OS rates in the HSS and HA group were 98% and 98% (*p* = 1.00), 79% and 88% (*p* = 0.50), and 60% and 64% (*p* = 0.61), respectively. Cox multivariate analysis showed that preoperative irregular anti-viral therapy, Child-Pugh grade B liver function, vascular invasion, and microvascular invasion (MVI) were independent risk factors for early postoperative RFS (within 2 years), and preoperative irregular anti-viral therapy and vascular invasion were independent risk factors for 5-year OS. ***Conclusions:*** Radical treatment of HCC combined with synchronous splenectomy, especially applicable to patients with Child-Pugh grade A liver function, can significantly improve early postoperative RFS in patients with stage T1 HCC and liver-cirrhosis-associated portal hypertension, but fail to improve OS.

## 1. Introduction

Hepatocellular carcinoma (HCC) is the fourth most common cancer and the second leading cause of cancer-related death in China [1]. Patients with HCC commonly have varying degree of cirrhosis, portal hypertension (PH), and secondary splenomegaly. In patients with HCC accompanied by clinically significant liver-cirrhosis-associated PH, determining the optimal treatment strategy other than liver transplantation is difficult and requires a comprehensive consideration of oncological features and liver function compensation status. PH and hypersplenism preclude some aggressive therapies, such as curative resection, local ablation, and TACE, for HCC due to pancytopenia, hyperbilirubinemia and hypoproteinemia, which are equally important negative factors for the adjuvant therapy or systemic treatments of recurrent liver tumor [2,3,4]. According to the European and American Association for the Study of Liver Diseases guidelines, patients with clinical manifestations of PH, such as gastrointestinal bleeding and ascites, are not suitable for routine hepatic resection. However, according to the latest Chinese guidelines for the treatment of HCC [5], delicate assessment of PH degree can help screening cases that are suitable for conventional surgical resection.

Until now, whether additional surgical procedures, such as synchronous splenectomy (or esophagogastric devascularization), could provide a better survival benefit is not clear. Some studies [6,7] have noted that synchronous splenectomy could upgrade surgical safety, improve liver function in Child-Pugh (CTP) class B, prolong RFS and OS, and also provide more alternative options for subsequent treatment. However, other studies have demonstrated [8,9] that splenectomy did not prolong OS in patients with HCC and PH. The definite causal link between improvement in liver function and a decreased tumor recurrence trend still remains controversial. Therefore, in this retrospective study, we aimed to investigate whether radical treatment of HCC combined with synchronous splenectomy could improve the survival benefit in patients with HCC and PH, compared with patients receiving radical treatment alone.

## 2. Material and Methods

### 2.1. Patients

In this study, 96 consecutively accrued HCC patients with liver cirrhosis and secondary PH underwent radical treatments (liver resection or radiofrequency ablation) and synchronous splenectomy (HS group) at Beijing Youan Hospital, Capital Medical University, from March 2011 to May 2019. As such, 48 patients from the HS group at stage T1 were defined as the HSS group. Another cohort of 42 patients with PH at stage T1 HCC who received hepatectomy (or radiofrequency ablation) alone during the same period served as the control group, defined as the HA group. All cases were followed up until March 2022. Indications for splenectomy included: splenomegaly with decreased leukocytes and platelets [10], demand for anti-hepatitis C virus therapy, history of gastrointestinal bleeding, and endoscopic significant esophagogastric varices with a positive red sign which was an additional indication for Hassab’s surgery (esophagogastric devascularization). The diagnosis of HCC was confirmed by histological examination or at least 2 radiologic images showing characteristic features of HCC, or persistent raised α-fetoprotein (AFP) >400 ng/mL together with 1 radiologic image showing characteristic features of HCC.

Radiofrequency ablation was performed in patients whose indocyanine green retention rate at 15 min (ICG-R15) >30%. Patients with stage T4 tumor, CTP grade C liver function, and splenic tumors or splenic abscess were excluded.

The study was conducted in accordance with the Declaration of Helsinki (as revised in 2013). All patients signed an informed consent form for the procedure approved by the Institutional Ethics Committee of the Beijing Youan hospital affiliated to Capital Medical University.

### 2.2. Preoperative Assessment

Clinical and biochemical features were collected for all patients: gender, age, etiology, AFP, total bilirubin, albumin, prothrombin time activity (PTA), history of tumor treatment, and oncological features. The CTP score was obtained according to the described classification proposed by Pugh et al. Liver ultrasound elastography was performed to assess liver stiffness [11].

Abdominal ultrasonography was performed routinely to obtain necessary information about the tumor. Additional contrast-enhanced computed tomography (CT) and/or contrast-enhanced magnetic resonance imaging (MRI) were performed to confirm the diagnosis of HCC and the extent of vascular invasion (peripheral branch vessel invasion or trunk branch invasion).

### 2.3. Surgical Procedure and Intraoperative Management

All operations were carried out under general anesthesia and performed in open and laparoscopic hepatectomy, or ablation combined with or without splenectomy. The splenectomy or Hassab‘s surgery was generally performed before hepatectomy. For splenectomy, the splenic artery trunk at the superior margin of pancreas was ligated first, followed by the splenic pedicle and the surrounding ligaments. After splenectomy and disconnection of the posterior gastric varices, the devascularization was continued upward from the gastric angle on the side of the stomach lesser curvature, and the gastric, esophageal branches of the coronary vein (including the perforating vessels) as well as high esophageal branches were disconnected using ultrasonic scalpel until 6–8 cm of the lower esophagus.

Intraoperative ultrasound was used to detect additional suspicious tumor nodules or direct the local ablation pathway. Parenchymal transection was completed by the ultrasonic scalpel and bipolar coagulation. Hepatic vessels <2 mm were coagulated with an ultrasound scalpel, whereas larger vessels and branched bile ducts were clipped after ligation. The Pringle maneuver was not usually adopted in light of decreased portal pressure secondary to first-step splenectomy. For local ablation, the RFA needle was inserted into the center of the target HCC nodule. RFA was applied continuously for 6–12 min. The RFA extent was determined mainly according to ultrasonography of the tumor covered by the hyperechoic ablated scope.

Anatomical resection was defined as removing anatomical Couinaud segments or whole hepatic lobe, and local hepatectomy as excision along the periphery of the tumor. Blood loss and operative duration were recorded.

### 2.4. Postoperative Treatment

Intravenous antibiotics were applied to prevent infectious complications. In the HS group or HSS group, if there was no evidence of active bleeding, anticoagulation therapy was started by subcutaneous injection of low-molecular-weight heparin (LMWH) to prevent the portal vein thrombosis (PVT) within 3 days after operation. Bedside blood flow ultrasonography was routinely performed to detect the presence of PVT.

All patients obtained R0 margin. Gross tissue paraffin specimens were pathologically confirmed with or without microvascular invasion (MVI). Postoperative mortality was defined as death within 30 days after surgery.

### 2.5. Follow-Up

Regular postoperative follow-up including outpatient and telephone method was carried out to record the recurrence date and survival time. RFS was defined from the operation date to the detection of tumor recurrence date. OS was defined as the interval between the date of operation and the date of tumor-related death. All patients were followed every 3 to 6 months, consisting of liver function biochemical test, serum AFP level, and liver ultrasonography or CT. The mean follow-up time after surgery was 75 months (range 35–135 months). When recurrence occurred within 1 year after resection, repeated liver resection was performed indicating that the liver function permitted. For recurrence occurring beyond 1 year after resection or inappropriate for reoperation, radiofrequency ablation, ethanol injection, or TACE was performed.

Persistent antiviral therapy with entecavir or tenofovir disoproxil was recommended for HBV-associated liver cirrhotic patients both preoperatively and postoperatively. In addition, prophylactic anticoagulation therapy with aspirin and warfarin to prevent PVT was extended to 3–12 months after discharge.

### 2.6. Statistical Analysis

All statistical analyses were performed with SPSS version 24.0 (IBM, Armonk, New York, NY, USA) and R language (version 3.5.2). Continuous variables were expressed as mean and standard deviation or median and interquartile range (IQR). The differences between groups were compared by the independent-samples *t* test or Mann–Whitney test. Categorical variables were reported as number of cases and prevalence, and the differences between groups were compared by the chi-square test or Fisher’s test. Patients’ survival curves were drawn using the Kaplan–Meier method and compared using the log-rank test. The multivariate analysis was performed using the Cox hazard proportional model to confirm the risk factors influencing RFS and OS. Two-sided *p* < 0.05 was considered statistically significant.

## 3. Results

During the study period, the clinical data of 106 patients with HCC and PH who received radical treatment and synchronous splenectomy (or Hassab’s surgery) were retrieved. For this study, 10 patients were excluded: 5 patients lost to follow-up, 2 patients presented with macrovascular invasion, 1 patient with splenic tumor, 1 patient with splenic abscess, and 1 patient with T4 stage (local peritoneal invasion). Finally, 96 consecutive patients were included in this study (HS group, Table 1). As such, 48 patients with HCC T1 stage from the HS group were defined as the HSS group (solitary tumor). The clinical data of 42 patients (HA group) from a pool of 233 patients with HCC T1 stage were collected, who met the PH criteria and received radical treatment alone.

### 3.1. Preoperative Characteristics and Intraoperative Data

The clinical characteristics of the three groups are detailed in Table 1 and Table 2. First, 58% of patients in the HS group received regular antiviral therapy before admission, 58% underwent liver resection, and 42% underwent radiofrequency ablation; 45% had a history of preoperative varices hemorrhage, while 56% underwent concomitant Hassab’s surgery. All five patients who died early postoperatively had CTP grade B liver function: one patient died of liver failure, two died of severe sepsis (one liver abscess and one abdominal infection), and two died of uncontrollable abdominal bleeding.

Median serum AFP was 9.9 ng/mL and 49.5 ng/mL in the HSS group and HA group, respectively. The median fibroscan was 18.8 Kpa and 14.5 Kpa in the HSS and HA group, respectively. The number of patients with CTP B liver function classification was nine (18%) and four (10%) in the HSS and HA group, respectively. The median size of the tumor nodule was 19.5 mm (IQR 13.8–30) and 20.0 mm (IQR 15–25) in the HSS and HA group, respectively.

The HSS group had a higher proportion of history of varices hemorrhage, lower hemoglobin, and worse coagulation mechanisms compared with the HA group. There were also statistically significant differences in intraoperative blood loss and operative duration between the two groups (*p* < 0.05). There was no early postoperative death in either group.

### 3.2. Survival Outcomes: RFS and OS

The mean median RFS of 22.3 months and mean median OS of 46 months were observed in the HS group during follow-up. For the HS group, RFS and OS were significantly different for different tumor stages (*p* < 0.01), demonstrating the best survival benefit for stage T1 HCC (Figure 1). Given the limited survival benefits for T2 or more advanced stage HCC, we did not set up an additional control cohort for tumor beyond stage T2. Furthermore, we compared the differences in RFS and OS according to different CTP scores and grades, which suggested that patients with grade A liver function had better RFS and OS at each time point than that with grade B liver function (*p* < 0.05) (Appendix A).

The median RFS during the whole follow-up in the HSS and HA group was 38.2 and 32.3 months (*p* > 0.05), with OS 59.4 and 54 months (*p* > 0.05), respectively. The RFS rates at 1, 2, 3, and 5 years in the HSS and HA groups were 95% and 81% (*p* = 0.041), 81% and 67% (*p* = 0.05), 64% and 62% (*p* = 1.00), 29% and 45% (*p* = 0.10) (Figure 2). The OS rates at 1, 3, and 5 years in the HSS and HA group were 98% and 98% (*p* = 1.00), 79% and 88% (*p* = 0.50), and 60% and 64% (*p* = 0.61), respectively. The OS rate during the whole follow-up was 79% and 64% in the HSS and HA group (*p* = 0.181), respectively (Figure 3).

### 3.3. Prognostic Factors

Through analysis of the overall sample (138 patients), Cox multivariable regression analysis showed that preoperative anti-viral therapy (HR 0.553, 95% CI 0.329–0.930, *p* = 0.026), CTP grade B liver function (HR 2.930, 95% CI 1.635–5.252, *p* = 0.000), vascular invasion (HR 2.561, 95% CI 1.467–4.471, *p* = 0.001), and MVI (HR 2.276, 95% CI 1.271–4.075, *p* = 0.006) were independent risk factors for 2-year RFS. Preoperative anti-viral therapy (HR 0.473, 95% CI 0.275–0.812, *p* = 0.007) and vascular invasion (HR 2.307, 95% CI 1.320–4.031, *p* = 0.003) were independent risk factors for worse 5-year overall survival (Table 3).

## 4. Discussion

PH coupled with hypersplenism not only impairs treatment options for HCC, but also reduces surgical outcomes and survival expectations. Perioperative bleeding and postoperative liver failure are the major concerns for cirrhotic patients with HCC and PH. According to the Barcelona liver cancer staging system recommendations, PH is a contraindication to routine liver resection, and the preferred optimal treatment is liver transplantation. However, with the advancement of surgical techniques and critical care management, some selected HCC patients with PH may still achieve long-term survival through conventional therapeutic surgery. As early as 1989, Takayama et al. [12,13] reported the advantage of synchronous or metachronous splenectomy for liver function improvement, as well as the increase in surgical safety. Therefore, prior experiences and some high-quality studies suggest PH should not be considered as an absolute contraindication for surgery, but rather as a standard preoperative assessment indicator [14]. Our data from a longitudinal cohort and a cross-sectional comparative cohort confirmed the survival advantage of synchronous splenectomy (or esophagogastric devascularization) in the patients with HCC, and in particular, significantly improved early RFS in stage T1 HCC, which is consistent with the findings of previous studies [6,9,15].

Our study provided a competitive alternative that some selected HCC patients with decompensated cirrhosis could benefit from, with simultaneous splenectomy and radical treatments. However, the postoperative mortality in the HS group (5.2%) was relatively high compared with other studies; all five patients who died early after operation had CTP grade B liver function, suggesting that simultaneous surgery would be safer and more beneficial if patients with CTP B liver function receive much more liver support before surgery. Most of the patients in our study had severe cirrhosis and esophagogastric varices; 45% patients of the HS group had a history of variceal bleeding. Unlike Western countries, where endoscopic therapy (sclerotherapy or varices ligation) and transjugular intrahepatic portosystemic shunt (TIPS) are the standard treatments for variceal patients, splenectomy combined with Hassab’s operation is more common in China, mainly due to the lack of organ donors and no better alternatives. Of the 54 patients who underwent the Hassab’s procedure in the HS group, 37 (68.5%) patients maintained stably and did not have variceal rebleeding during long-term follow-up, which is a satisfactory hemostatic outcome. Notably, although there was no early postoperative mortality or liver failure in either group, the HSS group showed significant intraoperative blood loss and operative duration compared to the HA group. This might be due to the more sophisticated Hassab’s surgery procedure.

Splenectomy can alleviate PH, elevate platelet count level, and ameliorate hyperbilirubinemia, which, in turn, promote hepatocyte regeneration [16]. In addition, a significant decrease in suppressive regulatory T cells and myeloid-derived suppressor cells after splenectomy is observed [17], and an increase in the number and function of lymphocytes, especially effector T cells, reverses the body’s immune suppression of the tumor and induces tumor regression, which may have a preventive effect on HCC recurrence after hepatectomy or radiofrequency ablation [18,19]. Despite the existence of the anti-tumor immune advantage, the duration of maintenance of this beneficial change in immune function in patients with cirrhosis is also a topic that deserves in-depth investigation. According to our two comparative cohorts (HSS group and HA group), splenectomy was shown to improve early postoperative RFS, but there was no significant advantage for longer RFS and there was no significant difference in overall OS between the two groups. Two recent meta-analyses [20,21] showed that simultaneous splenectomy prolonged both RFS and 5-year OS, but Xie et al. [8] concluded that simultaneous splenectomy did not significantly improve the OS after enrolling updated data from the literature. Thus, non-oncological factors, such as spleen size, degree of cirrhotic decompensation, and gender, which influence the duration of immunological advantage after splenectomy, should be considered.

CTP score is an important factor affecting RFS. After analyzing the HS group and the entire sample separately, patients with grade A liver function had better short- and long-term RFS than grade B. Some studies [22,23] pointed out that selected patients with CTP score 7 (grade B) liver function may still benefit from combined surgery, provided that they have a solitary and small tumor (less than 3 cm). Additionally, unfavorable factors affecting tumor recurrence and long-term survival include tumor vascular invasion. In this study, 37 patients in the HS group had vascular invasion by CT or MRI, 30 of which (81.1%) were peripheral-type branch vascular invasion, which was not consistent with the detection rate of MVI from the postoperative pathology [24,25], reminding us that preoperative enhanced vascular imaging may be more valuable than MVI in predicting long-term survival.

There are several potential limitations to this study. First, given that it was a retrospective study, selection bias may be inherent. To reduce this bias, we selected contemporary consecutive case control to support the conclusion and performed the multivariate analysis for adjustment of some confounding factors. Second, the sample size was relatively small. No control group comprising patients in stage T2 HCC without splenectomy may also limit the wider conclusion. It is necessary to enlarge the sample size and conduct a randomized control study to confirm the role of HS in improving the survival benefits in patients with HCC and PH in the future. In addition, the ethnicity of the subjects could also add to the selection bias as this study only evaluated Chinese patients.

## 5. Conclusions

Radical treatments of HCC combined with simultaneous splenectomy, especially for patients with CTP grade A liver function, can significantly improve early postoperative RFS in stage T1 HCC patients with liver-cirrhosis-associated PH. It is necessary to strengthen the assessment of the immunological function in long-term survival patients to further clarify the advantages of this joint therapy.

## Figures and Tables

**Figure 1 cancers-14-03155-f001:**
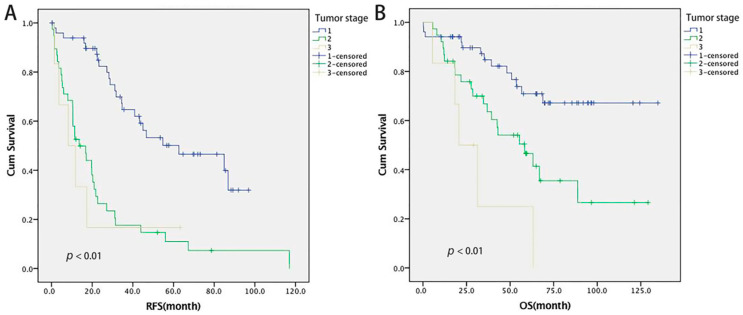
The RFS and OS curves for different HCC stage in the HS group (**A**: RFS; **B**: OS).

**Figure 2 cancers-14-03155-f002:**
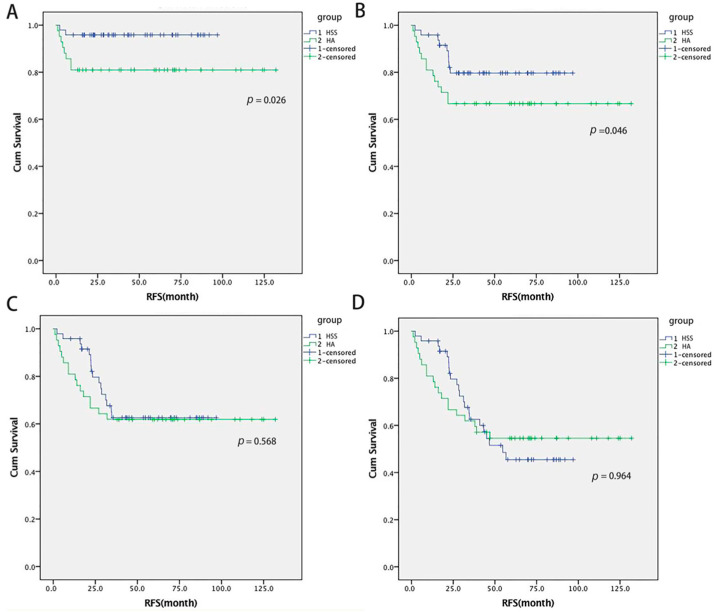
The RFS curves at different time points compared between the HSS group and HA group (**A**: 1-year RFS; **B**: 2-year RFS; **C**: 3-year RFS; **D**: 5-year RFS).

**Figure 3 cancers-14-03155-f003:**
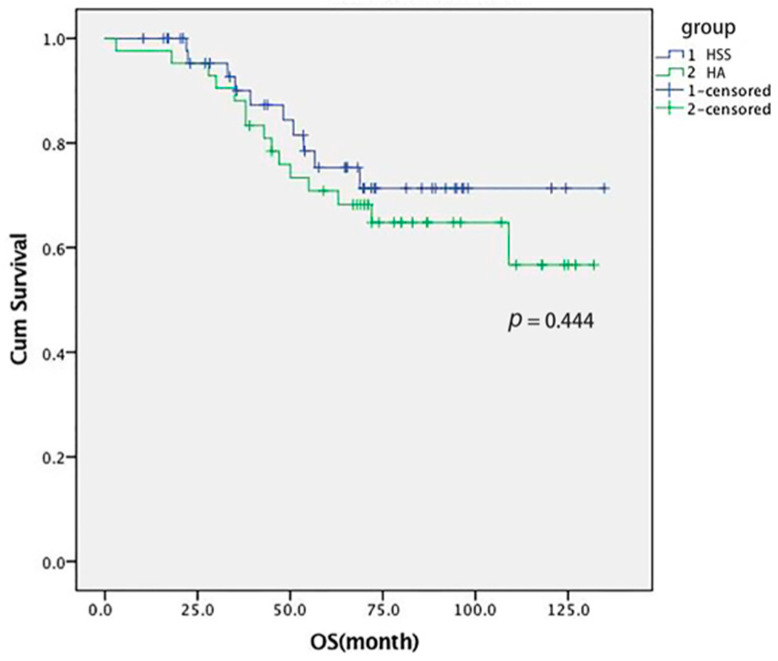
The OS curves compared between the HSS group and HA group.

**Table 1 cancers-14-03155-t001:** Baseline characteristics of the HS group population.

Variables	HS (*n* = 96)
Gender, male/female, *n* (%)	58 (60)/38 (40)
Age(year), Mean ± SD	54 ± 9
Etiology, HBV/HCV/alcohol or others, *n* (%)	76 (79)/14 (15)/6 (6)
History of HCC treatment, yes/no, *n* (%)	22 (23)/74 (77)
Preoperative anti-viral therapy, yes/no, *n* (%)	56 (58)/40 (42)
Varices hemorrhage, yes/no, *n* (%)	43 (45)/53 (55)
WBC (×10^9^/L), Median (IQR)	2.3 (1.8, 3.4)
HGB (g/L), Median (IQR)	114.5 (88, 134.2)
PLT (×10^9^/L), Median (IQR)	47 (37, 65.2)
ALT (U/L), Median (IQR)	28.5 (19, 41)
Tbil (umol/L), Median (IQR)	21 (15.3, 28.7)
Albumin (g/L), Mean ± SD	36.1 ± 5.4
Child-Pugh score, *n* (%)	
5	33 (34)
6	36 (38)
7	17 (18)
8	8 (8)
9	2 (2)
AFP (ng/mL), Median (IQR)	23.7 (3.4, 234.2)
PTA (%), Mean ± SD	72.5 ± 12.7
Fibroscan (Kpa), Median (IQR)	20.9 (14.5, 27.2)
Tumor diameter (mm), Median (IQR)	24 (14, 39.2)
Tumor number, 1/2/3, *n* (%)	82 (85)/10 (10)/4 (4)
Tumor stage, 1/2/3, *n* (%)	48 (50)/39 (41)/9 (9)
Vascular invasion, yes/no, *n* (%)	37 (39)/59 (61)
Pathological differentiation, high/middle/low, *n* (%)	24 (25)/35 (36)/37 (39)
MVI, yes/no, *n* (%)	23 (24)/73 (76)
Technique, resection/ablation, *n* (%)	56 (58)/40 (42)
Hassab’s surgery, yes/no, *n* (%)	54 (56)/42 (44)
Blood loss (ml), Median (IQR)	350 (200, 700)
Surgical duration (h), Median (IQR)	4.7 (3.6, 5.8)
Postoperative hospital stay (d), Median (IQR)	14.6 (11.1, 19.6)
Postoperative mortality, *n* (%)	5 (5)
Varices re-hemorrhage during follow-up, *n* (%)	25 (26)
Tumor recurrence during follow-up, *n* (%)	62 (65)
Overall RFS during follow-up (month), Median (IQR)	22.3 (10.5, 48)
1-year RFS, *n* (%)	67 (70)
2-year RFS, *n* (%)	46 (48)
3-year RFS, *n* (%)	31 (36)
5-year RFS, *n* (%)	18 (22)
OS(month) during follow-up, Median (IQR)	46 (22, 68.4)
1-year OS, *n* (%)	86 (90)
3-year OS, *n* (%)	56 (70)
5-year OS, *n* (%)	34 (49)

Abbreviations: WBC, white blood cell count; HGB, hemoglobin; PLT, platelet count; ALT, alanine transaminase; Tbil, total bilirubin; PTA, Prothrombin activity; MVI, microvascular invasion; RFS, recurrence-free survival; OS, overall survival.

**Table 2 cancers-14-03155-t002:** Baseline characteristics and perioperative data of stage T1 HCC treated with radical treatment with synchronous splenectomy (HSS group) compared with radical treatment alone (HA group).

Variable	HSS (*n* = 48)	HA (*n* = 42)	*p*-Value
Gender, *n* (%)			0.357
male	24 (50)	26 (62)	
female	24 (50)	16 (38)	
Age(year), Mean ± SD	52.6 ± 9.6	54 ± 7.9	0.438
Etiology			0.054
HBV	43 (90)	30 (71)	
HCV	5 (10)	10 (24)	
Alcohol or others	0 (0)	2 (5)	
Varices hemorrhage, *n* (%)	20 (42)	2 (5)	<0.001
History of HCC treatment, *n* (%)	10 (21)	7 (17)	0.815
Preoperative anti-viral therapy, *n* (%)	36 (75)	31 (74)	1.000
WBC (×10^9^/L), Median (IQR)	2.2 (1.8, 3.1)	2.4 (1.9, 2.9)	0.859
HGB (g/L), Median (IQR)	115 (86.2, 135.2)	126 (117, 137)	0.035
PLT (×10^9^/L), Median (IQR)	43 (36.8, 62)	44.5 (35.2, 56.2)	0.577
ALT (U/L), Median (IQR)	29 (15.8, 38)	30.6 (22.4, 45.3)	0.155
Tbil (umol/L), Median (IQR)	18.4 (14.8, 27.2)	19.3 (16.4, 25.4)	0.894
Albumin (g/L), Mean ± SD	36.8 ± 5	37.5 ± 3.9	0.417
Child-Pugh score, *n* (%)			0.108
5	20 (42)	27 (64)	
6	19 (40)	11 (26)	
7	6 (12)	4 (10)	
8	3 (6)	0 (0)	
AFP (ng/mL), Median (IQR)	9.9 (2.6, 135.7)	49.5 (7.8, 204)	0.05
PTA (%), Median (IQR)	72.5 (67.8, 80.2)	77.5 (72, 85)	0.006
Fibroscan (Kpa), Median (IQR)	18.8 (13.9, 25.4)	14.5 (11.6, 21.7)	0.018
Tumor diameter (mm), Median (IQR)	19.5 (13.8, 30)	20 (15, 25)	0.884
Technique, *n* (%)			0.327
resection	30 (62)	21 (50)	
ablation	18 (38)	21 (50)	
Extent of resection, *n* (%)			0.341
local	16 (53)	14 (67)	
anatomical	14 (47)	7 (33)	
Surgical margin (cm), Median (IQR)	0.5 (0.3, 1)	0.4 (0.2, 0.9)	0.753
Pathological differentiation			0.075
high	18 (38)	7 (17)	
middle	15 (31)	20 (48)	
low	15 (31)	15 (36)	
MVI, *n* (%)			0.777
no	39 (81)	36 (86)	
yes	9 (19)	6 (14)	
Hassab’s surgery, *n* (%)	28 (58)	0 (0)	
Blood loss (ml), Median (IQR)	300 (150, 425)	150 (100, 200)	0.002
Surgical duration (h), Median (IQR)	4.1 (3.2, 5.5)	3.5 (2.7, 4.3)	0.005
Postoperative hospital stay (d), Median (IQR)	15 (11.5, 17.6)	13.6 (9.7, 16.9)	0.127
Varices re-hemorrhage during follow-up, *n* (%)	12 (25)	4 (10)	0.101
PVT during follow-up, *n* (%)	12 (25)	3 (7)	0.055
Lethal PVT during follow-up, *n* (%)	5 (10)	1 (2)	0.212

Abbreviations: WBC, white blood cell count; HGB, hemoglobin; PLT, platelet count; ALT, alanine transaminase; Tbil, total bilirubin; PTA, Prothrombin activity; MVI, microvascular invasion; PVT, portal vein thrombosis.

**Table 3 cancers-14-03155-t003:** Multivariate analyses of prognostic factors for 2-year RFS and 5-year OS.

Variables	2-Year RFS			5-Year OS		
	HR	95% C. I	*p*-Value	HR	95% C. I	*p*-Value
Anti-viral therapy (yes vs. no)	0.553	0.329–0.930	0.026	0.473	0.275–0.812	0.007
Child-Pugh grade (B vs. A)	2.930	1.635–5.252	0.000			
Vascular invasion (yes vs. no)	2.561	1.467–4.471	0.001	2.307	1.320–4.031	0.003
MVI (yes vs. no)	2.276	1.271–4.075	0.006			

Abbreviations: MVI, microvascular invasion.

## Data Availability

Data will be available upon request to the corresponding author.

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
