# Peer review of "The Benefits of Radical Treatments with Synchronous Splenectomy for Patients with Hepatocellular Carcinoma and Portal Hypertension"

_cancers, 2022, doi:10.3390/cancers14133155_

Round 1
Reviewer 1 Report
Dear authors,
I have read with interest this manuscript that concerns the overall interesting topic of hepatectomy plus splenectomy in HCC patients with hypersplenism. Please be advised that some changes should be done as follows:
1. To may knowledge the first references on this argument are some from Japan, such as Takayama T et al. The role of splenectomy in patients with hepatocellular carcinoma and hypersplenism as an aid to hepatectomy. Nihon Geka Gakkai Zasshi. 1989 and Sugawara Y et al. Splenectomy in patients with hepatocellular carcinoma and hypersplenism. J Am Coll Surg. 2000. This references are missing and should be added together with all the pertinent literature.
2. It is unclear the numerosity of the included patients. 96 or 138? If 48 HSS (part of 96) were compared with 42 HA, what about the remaining 48 of the first 96?
3. What HA stands for? The explanation is missing.
4. Please add a paragraph named definitions to include of the adcopted definitions. Such as the definion for major resection, which usually is based on the Brisbane classification (>3 segments and not >2).
5. The follow-up was closed in March 2022. I do not think that this is appropriate because some patients do not have enough f/up.
6. While the performance of the Cox on the whole population is correct (138=96+42 - although the remaining 48 patients from the 96 cohort are not presented - please see point number 2) I believe that it might of interest to have also the Cox on the two compared groups HSS and HA.
7. LMWH were performed to prevent PVT. Do you mean thrombosis of the splenic vein in the HSS group? Anyway, these data are not shown.
8. Some patients in the HSS group had also the Hassab operation. This might explain the increased in blood loss. Please add some sentences on this procedure and on its results.
9. Surgical technique for the liver, the spleen, the Hassab procedure and RF should be added.
10. Ideally, a better comparation btw HSS and HA would require a kind of statistatical gimmick such as the usual propensity matching or the inverse propability weighting.
Reviewer 2 Report
Hepatocellular carcinoma (HCC) is the most common type of primary liver cancer accounting for more than 80% of all cases. Early stage HCC (Barcelona Clinic Liver Cancer-BCLC 0/A) could benefit from potentially curative treatments, such as liver resection, liver transplantation or local ablative therapies (radiofrequency ablation and microwave ablation); unfortunately, only 30-40% of patients present early-stage HCC (BCLC 0-A) at diagnosis and recurrence rate is high also due to the absence of approved adjuvant therapies. For intermediate (BCLC B) and advanced-stage HCC (BCLC C), treatment has a palliative intent and it includes local therapies, such as transarterial chemoembolization (TACE) and 90Yttrium transarterial radioembolization, or systemic therapy However, HCC arise often in the context of an underlying liver disease which limits therapeutic options and raise the issue of tolerability of systemic treatments.
The study addresses a very timely and important topic in this setting.
Some changes are required in my opinion:
- a linguistic revision by a professional service is highly suggested, since there are some minor overisights and mistakes to be corrected
- the medical treatment scenario for HCC patients should be better explained in the introduction section, and some recent papers added, only for a matter of consistency (PMID: 34429006; PMID: 29968763 ; PMID: 34167433)
- the limitations of the current study should be further highlighted, since a major issue is represented by the small sample size and the selection bias due to ethnicity. Please revise accordingly.
Author Response
- a linguistic revision by a professional service is highly suggested, since there are some minor overisights and mistakes to be corrected
-Response 1: Some oversights and mistakes have been corrected with the help of linguistic professionals. We prefer to language services at https://www.mdpi.com/authors/english if the revised version can not meet the linguistic requirements of the editorial board.
- the medical treatment scenario for HCC patients should be better explained in the introduction section, and some recent papers added, only for a matter of consistency (PMID: 34429006; PMID: 29968763 ; PMID: 34167433)
-Response 2: Done.
- the limitations of the current study should be further highlighted, since a major issue is represented by the small sample size and the selection bias due to ethnicity. Please revise accordingly.
-Response 3: Done.
Round 2
Reviewer 1 Report
I thank the authors for their efforts in working on their manuscript according with my suggestions. I believe that the manuscript has been improved, but please be advised that there are some more point to fix:
Response 3: In the “Abstract” and “Materian and Mehods” section, we depicted the meaning of HA, that is who received HCC radical treatment alone at stage T1 (HA group).
Well, this is NOT the correct way of using an abbreviation or an acronym. HCC is already an abbreviation that cannot be further abbreviated as H; more importantly the "A" is not attached to the H so the reader really may find some difficulties in understanding the meaning.
-Response 4: In view of the severe cirrhosis and inadequate liver function reserve, extensive liver resection strategies were less commonly used, and thus, we did not strictly adopt Brisbane's definition. Instead, >2 segments were labeled as “major”, which is only a personal definition.
Please write down some where in the manuscript this consideration, which anyway is personal. Be advised that the use of more standard classification, nomenclature and etc... helps the reader to understand the shown data.
here were cases that remained event-free at the end of the follow- up point and we call them right-censored. Currently statistical softwares (SPSS or R) can handle right-censored cases very well.
The follow-up is too short and there is no statistical software automatic system to fix this. Those patients with too short f/up should be deleted. Nobody can anticipate the course of a given disease without a minimum f/up, that in surgical oncology should be at least 12 months.
-Response 7: LMWH was used to prevent PVT (portal vein thrombosis, not splenic vein thrombosis). Splenic vein thrombosis alone does not generally have an adverse prognostic impact, so we did not record the incidence of splenic vein thrombosis.
This is interesting but unusual. In my knowledge (I am a liver surgeon) there is no common use of LMWH to prevent postoperative PVT. A different situation might be the case of transplantation, but this is not your case. Please add some explanations or references.
Reviewer 2 Report
The authors addressed all the issues we raised.
We recommend Acceptance.
Author Response
Thank you for your comments and recognition of our study.